# Unveiling the Hidden Entropy in ZnFe_2_O_4_

**DOI:** 10.3390/ma15031198

**Published:** 2022-02-04

**Authors:** Miguel Angel Cobos, Antonio Hernando, José Francisco Marco, Inés Puente-Orench, José Antonio Jiménez, Irene Llorente, Asunción García-Escorial, Patricia de la Presa

**Affiliations:** 1Instituto de Magnetismo Aplicado (UCM-ADIF), CSIC, 28260 Las Rozas, Spain; micobos@ucm.es (M.A.C.); antherna@ucm.es (A.H.); 2Donostia International Physics Center, 20018 Gipuzkoa, Spain; 3IMDEA Nanociencia, 28049 Madrid, Spain; 4 Industrial Engineering Department, Universidad de Nebrija, 28015 Madrid, Spain; 5Instituto de Química Física Rocasolano, 28006 Madrid, Spain; jfmarco@iqfr.csic.es; 6Institut Laue-Langevin, CEDEX 09, 38042 Grenoble, France; puenteorench@ill.fr; 7Instituto de Ciencia de Materiales de Aragón (ICMA), CSIC, 50009 Zaragoza, Spain; 8Centro Nacional de Investigaciones Metalúrgicas (CENIM), CSIC, 28040 Madrid, Spain; jimenez@cenim.csic.es (J.A.J.); irene@cenim.csic.es (I.L.); age@cenim.csic.es (A.G.-E.); 9Department of Material Physics, Complutense University of Madrid, 28040 Madrid, Spain

**Keywords:** zinc ferrite, hidden entropy, spin disorder, neutron diffraction, calorimetry

## Abstract

The antiferromagnetic (AFM) transition of the normal ZnFe_2_O_4_ has been intensively investigated with results showing a lack of long-range order, spin frustrations, and a “hidden” entropy in the calorimetric properties for inversion degrees δ ≈ 0 or δ = 0. As δ drastically impacts the magnetic properties, it is logical to question how a δ value slightly different from zero can affect the magnetic properties. In this work, (Zn_1-δ_Fe_δ_)[Zn_δ_Fe_2-δ_]O_4_ with δ = 0.05 and δ = 0.27 have been investigated with calorimetry at different applied fields. It is shown that a δ value as small as 0.05 may affect 40% of the unit cells, which become locally ferrimagnetic (FiM) and coexists with AFM and spin disordered regions. The spin disorder disappears under an applied field of 1 T. Mossbauer spectroscopy confirms the presence of a volume fraction with a low hyperfine field that can be ascribed to these spin disordered regions. The volume fractions of the three magnetic phases estimated from entropy and hyperfine measurements are roughly coincident and correspond to approximately 1/3 for each of them. The “hidden” entropy is the zero point entropy different from 0. Consequently, the so-called “hidden” entropy can be ascribed to the frustrations of the spins at the interphase between the AFM-FiM phases due to having δ ≈ 0 instead of ideal δ = 0.

## 1. Introduction

Zinc ferrites belong to the spinel structure (Zn_1-δ_Fe_δ_)^A^[Zn_δ_Fe_2-δ_]^B^O_4_, where A and B represent, respectively, the tetrahedral and the octahedral sites of the cubic structure and δ is the inversion degree parameter. There is a general agreement in the scientific community that δ plays the fundamental role in the magnetic properties of (Zn_1-δ_Fe_δ_)^A^[Zn_δ_Fe_2-δ_]^B^O_4_ with δ ≠ 0 [1,2,3,4]. For δ = 0, the ferrite is denominated normal, and it has been predicted to have an antiferromagnetic (AFM) transition around 10 K [5]. For δ ≠ 0, the zinc ferrite becomes ferrimagnetic (FiM) with magnetization that depends on δ as M = δ·(5.9 µ_B_) at 5 K [4] and can reach high magnetization values even at room temperature.

Several articles have been published attempting to identify the AFM transition with a large variety of techniques like neutron powder diffraction (NPD), Mossbauer and µRS spectroscopy, magnetometry, calorimetry, among others [4,6,7,8,9,10,11,12,13,14,15,16,17,18]. Most of the results on NPD report on broad peaks that suggest lack of long-range order; Fe magnetic moment smaller than 5.9 µ_B_ at the octahedral sites; a positive Curie–Weiss temperature and a magnetic entropy which is half that of the theoretical one. This reduced entropy has been observed in other spinel structures [15,19,20], and has been described as a “hidden” entropy, an entropy different from 0 at the zero point entropy [19]. Since in most of the experimental works on normal spinel, it is assumed δ ≈ 0 [21,22,23,24], the main question is: how critical is it to have δ ≈ 0 instead δ = 0.

This work shows that a δ slightly different from zero (δ = 0.05) dramatically affects the magnetic properties. For this goal, two samples are investigated: one with δ = 0.05 (that can be considered as δ ≈ 0) and δ = 0.27 (δ ≠ 0). X-ray and neutron powder diffraction, calorimetry at various applied fields (0, 1, 5, and 9 T), and Mossbauer characterizations have been performed. The results show that for δ = 0.05, around 40% of the unit cells suffer a Zn/Fe cation exchange with a huge impact on the magnetic interactions inside the unit cell.

## 2. Materials and Methods

Different treatments were carried out in powder zinc ferrite supplied by Alfa–Aesar (Alfa Aesar GmbH & Co KG, Germany) (99% purity) in order to obtain samples with different inversion degrees. First, the sample was calcined for 2 h at 1100 °C with a slow cooling down at an average speed of 7 °C/min; the sample was named ZFO-0.05. Later, the commercial powder was milled up to 50 h in a Retsch PM4 planetary mill (Retsch GmbH, Haan, Germany), in a 250 cm^3^ capacity jar and 1 cm diameter balls stainless steel, at a rotating speed of 275 rpm. This milled sample was then subjected to an annealing temperature of 400 °C for 1 h in order to decrease the inversion degree. The sample name was ZFO-0.27.

The samples were initially characterized by X-ray Diffraction (XRD) at room temperature (RT) using Co radiation in a Bruker AXS D8 diffractometer equipped with a Goebel mirror and a LynxEye detector (Bruker AXDS, GmbH, Karlsruhe, Germany). XRD spectra were collected in Bragg-Brentano geometry over a 2θ range from 10° to 120° with a step of 0.01°. Additionally, the magnetic structure has been studied using neutron powder diffraction (NPD) recorded at RT and 2 K at the Institute Laue-Langevin, Grenoble (France) with the high-resolution D2B (λ = 1.59 Å) diffractometer [25,26,27]. Powder samples were placed in vanadium cans and mounted onto a standard cryostat. Full diffraction patterns were obtained with a step of 0.05° with an acquisition time adjusted to obtain sufficient quality diffraction profiles (optimal counting statistics).

The obtained XRD and NPD patterns were analyzed using TOPAS v6.0 (Bruker AXDS, GmbH, Karlsruhe, Germany). Rietveld refinements were performed using as starting points the standardized structure for zinc ferrite taken from Pearson crystallographic database [28]. The quality of the refinements was evaluated by the statistically expected least-squares factor (R_exp_), the weighted summation of residual of the least-squares fit (R_wp_), and the goodness of fit (GoF or chi-square, whose limit tends to 1) [29].

Mossbauer experimental data were recorded from the sample ZFO-0.05 at 298.0, 77.0, and 8.8 K in the transmission mode using a conventional constant acceleration spectrometer, a ^57^Co(Rh) source, and a He closed-cycle cryorefrigerator.

Heat capacity was measured by PPMS (Physical Properties Measurement System) of Quantum Design in a range of temperatures between 2 and 300 K at zero applied field and at H = 1, 5, and 9 T from 2 to 40 K for both samples. For each applied field, the magnetic entropy change was calculated as $\Delta S=\int_{2}^{40}C/TdT$.

## 3. Results and Discussion

XRD patterns of samples recorded at room temperature confirm the presence of a spinel structure that matches with the JCPDS file of Franklinite (JCPDS 22-1012), as observed in Figure 1. Rietveld refinements of the NPD and XRD data have provided microstructural parameters like cell lattice, inversion degree, O-position, crystallite size, and microdeformation shown in Table 1. As can be seen, these refinements have provided the same value of the inversion degree δ = 0.05 for the sample ZFO-0.05, whereas the inversion degree obtained for the ZFO-0.27 sample depends on the diffraction technique used. In spinel structures, the nuclear and magnetic diffraction peaks occur at the same scattering angles, giving rise to a strong correlation between the inversion parameter (occupancy) and magnetic moments. This correlation may lead to unsatisfactory fits and differences between the inversion parameter determined from the XRD and NPD patterns. To solve it, XRD and NDP data were simultaneously fitted by the Rietveld, constraining the microstructural parameters like inversion degree, the fractional coordinate of the oxygen atom, and particle size to the same value for the patterns recorded at both 2 and 300 K. The value of the lattice parameters at these temperatures were also constrained to be the same for both XRD and NPD patterns. Finally, independent magnetic moments were included in the refinement for the NPD profiles recorded at 300 and 2 K under the assumption that the cation distribution is the same at both temperatures. The best-fitted microstructural parameters obtained by the combined analysis lead to an inversion degree of 0.27, taken as the best value, for the good quality by low values of the R-factors and GoF (R_wp_ = 3.27, R_exp_ = 2.65, and GoF = 1.26).

It is worth noting that the crystallite size of sample ZFO-0.27 is 15 nm, below the magnetic critical size, whereas ZFO-0.05 has a crystallite size of micrometers, i.e., bulk size.

Figure 2 shows the NPD patterns for δ = 0.05 and 0.27 in the range 2θ = 5° to 37°. As the magnetic reflections superimposed the nuclear ones, an additional contribution in the intensity occurs for the peaks at (111) and (220) when some Fe^3+^ magnetic ions are also occupying A sites. In addition, peak broadening can occur by reducing the crystallite size and/or increasing the lattice strain. Besides differences in the height and shape of the diffraction peaks associated with these two factors, it can be observed a broad peak at a smaller angle for the sample with δ = 0.05 (Figure 2), corresponding to the (1 0 ½) reflection, which indicates the presence of a short-range order (SRO) antiferromagnetic (AFM) order. The area of this peak decreases with increasing δ. The broadness of the peak at (1 0 ½) for δ = 0.05 describes a picture of lack of AFM long-range order that has also been reported by other authors [9].

Figure 3 shows the heat capacity and the entropy increment for both samples. As can be seen in Figure 3A, *C/T* shows a sharp peak around 10 K for the sample with δ = 0.05, which is close to the Neel temperature of the normal zinc ferrite, whereas this peak becomes broader and shifts to lower temperatures when the inversion degree is δ = 0.27.

The total heat capacity (*C_T_*) between 2 and 40 K is the sum of both the magnetic (*C_m_*) and vibrational (*C_L_*) heat capacities, while at 40 K, there are only vibrational contributions.

The vibrational heat capacity is given by the following equation:(1)$$C_{L}(T)=\frac{12}{5}\pi^{4}R{\left(T/\theta_{D}\right)}^{3}$$
where *R* is the ideal gas constant, *θ_D_* is the Debye temperature, and *T* the temperature. By subtracting the vibrational contribution from the total heat capacity, the magnetic contribution to the entropy can be calculated by integrating the *C/T* curve.

The entropy increment is field-dependent in the case of very low inversion degree δ = 0.05 (i.e., δ ≈ 0). When magnetic fields from 1 to 5 T are applied, the entropy increment increases, being the increment almost independent of the field in this field range; however, the entropy decreases for an applied field of 9 T (Figure 3B). On the other hand, in the case of the larger inversion degree, δ = 0.27, the entropy increment is independent of the applied field (Figure 3C).

Figure 4 shows the total ΔS_T_, magnetic ΔS_m,_ and lattice ΔS_L_ contributions to the entropy for δ = 0.05 at *H* = 0 T, and Table 2 collects the values at 40 K at different applied fields. The entropy increment for the sample with δ = 0.27 is field-independent.

In the absence of an applied field, the theoretical value of the magnetic entropy increment from the spin ordered state at 0 K to the paramagnetic (PM) state at temperatures well above the magnetic transition is 2Rln(2J+1)=29.7 J/mol·K, where *J* is the total quantum moment, with *J* = 5/2 for Fe^3+^. The difference between the theoretical and the experimental value contains information about additional contributions to the entropy, and it is defined as the zero-point entropy at 0 K [19,20]. It is important to remark that for (Zn_1-δ_Fe_δ_)^A^[Zn_δ_Fe_2-δ_]^B^O_4_, the parameter δ remains constant in the whole temperature range for each sample. Consequently, the entropy changes at low temperatures must be ascribed to an increment in the magnetic contribution and, to a lesser extent, to the lattice vibrational effect.

When assuming δ = 0, the only exchange interaction is the B-B which is AFM. As a pair of Fe and Zn atoms interchange their crystal positions, there appears the AFM A-B super exchange interaction, which is significantly stronger than the B-B one. This AFM A-B interaction leads to the ferromagnetic (FM) ordering of the three remaining Fe^3+^ in the B site. Consequently, the first neighboring Fe^3+^ atoms in the B sites (with 4 Fe^3+^ per site) surrounding the A site occupied by a Fe^3+^ experience some kind of frustration due to the two competing AFM and FiM interactions. Figure 5 shows a scheme of this situation.

Inside a single unit cell (see Figure 5), the smallest local inversion (δ_c_) is either 0 or 1/8 = 0.125, the last one corresponding to the exchange of a single Zn/Fe pair in the unit cell. Assuming that only a single Zn/Fe exchange can occur in a cell, around 40% of the unit cells suffer a single Zn/Fe cations exchange when the macroscopic δ is as small as 0.05. Those unit cells with δ_c_ = 0.125 are FiM (40%) with a magnetic moment of 5.9 μ_B_, [4] whereas for δ_c_ = 0 the cells are AFM (60%). This pictures the dramatic effect that δ ≈ 0 can have over the magnetic and calorimetric properties.

(a) Inversion degree δ ≈ 0

The entropy increment at different applied fields of sample ZFO-0.05 is shown in Figure 3. At H = 0 and 40 K, the magnetic entropy increment is Δ*S_m_* = 8.7 J/mol·K, which can be associated with AFM to PM transition. Assuming that for δ=0.05 a 60% of the sample is AFM, the expected Δ*S_m_* is close to 0.6·2RLn(2J+1) ≈ 18 J/mol·K. Therefore, the small experimental value of 8.7 J/mol·K indicates that only 29% of the sample has evolved from AFM to the paramagnetic phase. In summary, instead of the expected 60%, only 29% of the sample seems to be AFM ordered.

The 71% of the sample volume that does not contribute to Δ*S_m_* could be ordered with transition temperature above 40 K or disordered in the whole 0–40 K range. It is normally associated with a hidden or missing entropy observed in other spinels [19,20].

By applying a magnetic field of 1 T, the corresponding Δ*S_m_* rises up to 17.1 J/mol·K, indicating that a volume fraction of 57% has evolved from 0 entropy to 29.7 J/mol·K. This 57% is close to the calculated AFM fraction of 60%. At 40 K, only contributions to the entropy of the lattice or a magnetically ordered phase are expected; therefore, both experimental curves, with and without applied field, are matched at this temperature (Figure 6). As can be seen, the effect of the field is shown to promote a decrease of the low-temperature magnetic entropy and also to raise its increasing rate nearby 0 K. According to this result, at zero applied field, a fraction of the spins seems to be disordered and thereby contributing in a small amount to the entropy increment, so giving rise to the “hidden entropy.” However, at low temperatures, the field gradually orders these disordered spins, contributing to the magnetic entropy increment when the temperature rises and the PM phase is achieved. In conclusion, the effect of the field allows us to unravel the hidden entropy origin as the spin disordered volume fraction vanishes.

In the experimental curves, two contributions can be distinguished: (a) The entropy increase associated with the AFM-PM transition Δ*S_m_*(0T, 40K) = 8.7 J/mol·K, which corresponds to a volume fraction of 29% of AFM state. (b) The contribution resulting from the spin disordered regions is field-dependent. The volume fraction of the spin disordered regions can be inferred by subtracting Δ*S_m_*(0T, 40K) = 8.7 J/mol·K to ΔS*_m_*(1T, 40K) = 17.1 J/mol·K, that leads to a spin disordered contribution of 8.4 J/mol·K, corresponding to a volume fraction of 28% of the spin disordered state. This volume fraction of spin disordered Fe^3+^ is expected to be located at the FiM-AFM interphase. The remaining 43% fraction is expected to have a FiM ordering with Curie temperature above the 40 K, [30]; therefore, this fraction does not contribute to the entropy increment.

The decrease of the entropy with the applied field is worth noting, from 17.1 J/mol·K at 1 T down to 14.2 J/mol·K at 9 T (Figure 3B and Table 2). This effect can be ascribed to the decrease of magnetic entropy on a PM system induced by an applied field.

Figure 7 shows the calculated Δ*S_m_*(µ_0_H,T) for H = 1 T and H = 9 T, according to the expression:(2)$$\;S_{m}\left(H,T\right)=0.56R\left(\left[ln\frac{\sinh\left(2J+1\right)x}{\sinh x}\right]-\left(2J+1\right)x\coth\left(2J+1\right)x+x\coth x\right)$$
with $x=\mu_{B}\mu_{0}H/k_{B}T$, where $\mu_{B}$ is the Bohr magneton and *µ_0_H* is the applied magnetic field.

Equation (2) accounts for the decrease of Δ*S_m_* (T) at 9 T with respect to Δ*S_m_* (T) at 1T observed experimentally (see Figure 2), as illustrated by Figure 7.

The 298 K Mossbauer’s spectrum (Figure 8) is composed by a PM quadrupole absorption, which was best-fitted to two quadrupole doublets with hyperfine parameters characteristic of high spin Fe^3+^ in octahedral oxygen coordination: δ_1_ = 0.34 mms^−1^, Δ_1_ = 0.26 mms^−1^ and δ_2_ = 0.34 mms^−1^, Δ_1_ = 0.57 mms^−1^. The 77 K spectrum still shows a paramagnetic doublet. The 8.8 K spectrum, however, shows a broad unresolved magnetic pattern (Figure 8). This spectrum, characteristic of a system experiencing magnetic relaxation, indicates that the measurement temperature is close to the magnetic ordering temperature of this particular zinc ferrite sample. As mentioned previously, it is known that the critical temperature of well-crystallized, canonical zinc ferrite (which should be a direct spinel) is close to 10 K and that this temperature increases if it is partially inverse [31]. Thus, the present result is compatible with a zinc ferrite sample having a very small inversion degree, as is the case.

Fitting the 8.8 K spectrum is a complicated matter. Since the sample is affected at such temperature by the relaxation induced by thermal fluctuations, it cannot be properly fitted using a static hyperfine magnetic field distribution (although this type of fit can be used as a first approximation) [32]. A more rigorous fit model should consider the occurrence of dynamical effects associated with the thermal fluctuations that strongly affect the line shape. In such a way, we can disentangle these dynamic effects from the possible existence of different configurations arising from different local environments (vide infra). Therefore, we fitted the 8.8 K spectrum following the approach described in ref. [33], which uses the Blume and Tjon line shape [34] to account for the occurrence of magnetic relaxation. The spectrum could not be satisfactorily fitted using only a unique configuration affected by thermal fluctuations as it should be the case if the sample was a direct spinel and only magnetic order arising from the Fe^3+^(B)-Fe^3+^(B) AFM exchange interaction was present. Instead, the spectrum was best-fitted using three different contributions characterized by hyperfine magnetic fields of 46.6 T, 37.2 T, and 26.7 T accounting for 34%, 29%, and 37% of the spectral area, respectively. We want to mention that because of the small inversion degree of the sample and the complexity of the 8.8 K spectrum, the isomer shifts of the various components fitted do not appear to be sensitive to differences in the Fe^3+^ cation coordination, being around δ = 0.50 mms^−1^ (i.e., mainly characteristic of octahedral sites) in the three cases.

Although in Mossbauer spectroscopy the type of magnetic order cannot be inferred without the application of an external magnetic field, we can plausibly associate, based on the entropy measurements, the components obtained from the fit of the 8.8 K Mossbauer spectrum to various regions in the solid having different magnetic order. As shown in Figure 8, the spectrum was best fitted to three different components: two relatively well-defined contributions and a very broad one. Following the reasoning above, these two well-resolved components could be associated with a ferromagnetic phase (higher hyperfine magnetic field, 46.6 T) and an AFM one (intermediate hyperfine magnetic field, 37.2 T). The broad unresolved contribution having the smaller hyperfine field (26.7 T) would correspond to a disordered spin phase due to the thermal fluctuations and local fluctuations of the super exchange interactions or structural disorder at the FiM-AFM interphase. This disordered phase accounts for 37% of the Mossbauer spectral area, a percentage that, considering the limitations of fitting a broad and complex spectrum, is reasonably close to the value obtained from the entropy measurements. The result is interesting since it confirms the statement formulated long ago that suggested that the partially inversed character of zinc ferrite, even in small proportions, makes the network of its magnetic interactions highly frustrated [31].

(b) Inversion degree δ = 0.27

For a high inversion degree, FiM cells percolate and are coupled to each other, giving rise to long-range FiM order; consequently, the presence of a large volume fraction of spin disordered atoms is not more possible. In this condition, the magnetization of the A sublattice is uniform in the whole crystal, and the mixture of magnetically coupled FiM and AFM cells gives rise to a single magnetic configuration. In other words, the long-range super exchange coupling lifts the ground state degeneracy, as just an applied magnetic field does.

In the case of δ = 0.27, this implies a large density growth of the FiM clusters that leads to a percolation among the FiM particles, reducing the volume of both AFM and spin disordered regions. Therefore, the entropy increase is insensitive to any applied magnetic field (from 1 to 9 T) (Figure 3C) with a value Δ*S_m_* = 5.5 J/mol·K. In this condition, the volume fraction of the sample contributing to Δ*S_m_* is 18%, much smaller than the corresponding sample with δ = 0.05, which was 57%. The presence of long-range order is confirmed by the macroscopic FiM associated with the decrease of the area enclosed by the halo of the NPD experiment that has almost vanished.

The plausible magnetic structure of the samples is represented in Figure 9. Their magnetic order evolution is interpreted in terms of the applied magnetic fields in the sample with δ ≈ 0 and of the percolation array when δ = 0.27.

## 4. Conclusions

Calorimetric study leads to understanding the so-called hidden entropy in samples with very low inversion degree. For a sample with δ = 0.05 (δ ≈ 0), a single pair of Zn-Fe cations have exchanged their places within the 40% of unit cells giving place to three magnetic regions: one of short-range order of FiM cluster, another one short-range order of AFM regions, and a third one in the interphase between AFM and FiM with frustrated or disordered magnetic moments. When 1 T magnetic field is applied, there is an increment of entropy with respect to H = 0 originated by the orientation of the disorder magnetic interphase lying between the FiM and AFM regions. However, for a field of 9 T, the magnetic entropy decreases because the high applied field orients all the magnetic moments more strongly at the PM high-temperature phase. The Mossbauer results show a complex spectrum with three main interactions with populations close to those determined by calorimetric experiments. Therefore, both characterizations confirm the presence of the three regions with AFM, FiM, and spin frustrations clusters with a population of around 1/3 each one. It is worth noting that the entropy is a state function, i.e., it is independent of the fluctuations of the spins near the magnetic transition observed by Mossbauer spectroscopy 8.8 K.

In the case of a high inversion degree (δ = 0.27), the magnetic entropy is invariant under the applied field, associated with the percolation of the FM cluster.

Furthermore, as a corollary, the great difficulty obtaining zinc ferrite with δ = 0 is verified because a very tiny amount of Fe^3+^ migrated in the A positions of the spinel produces a sufficient disturbance in the magnetic order.

## Figures and Tables

**Figure 1 materials-15-01198-f001:**
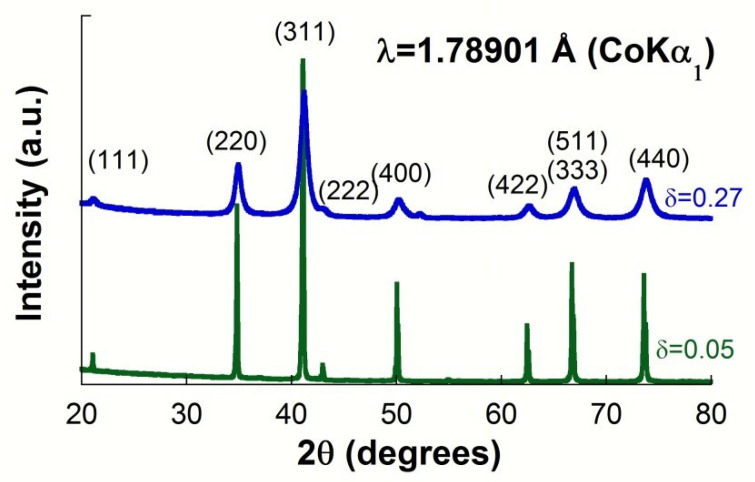
Diffraction patterns recorded at room temperature for the sample with δ = 0.27 and δ = 0.05.

**Figure 2 materials-15-01198-f002:**
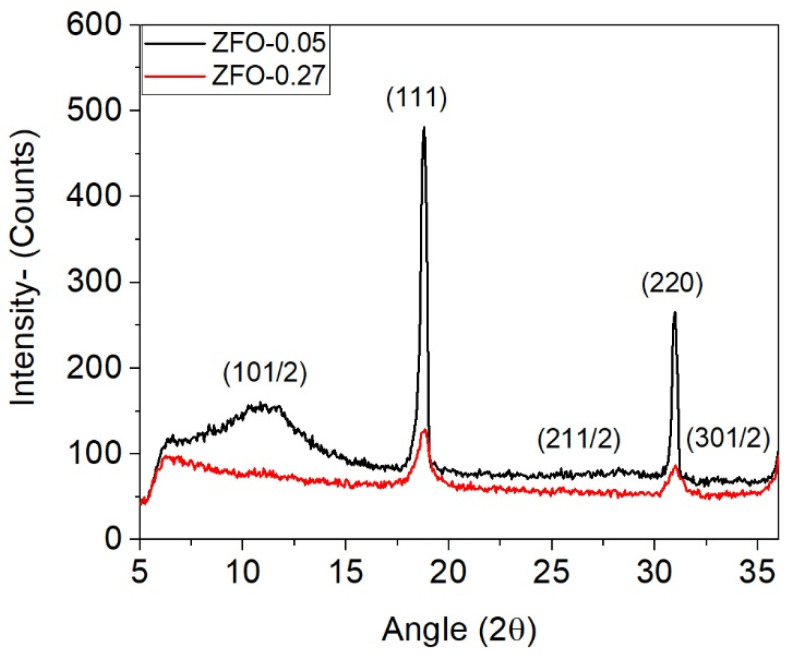
NPD patterns for samples with δ = 0.05 and δ = 0.27.

**Figure 3 materials-15-01198-f003:**
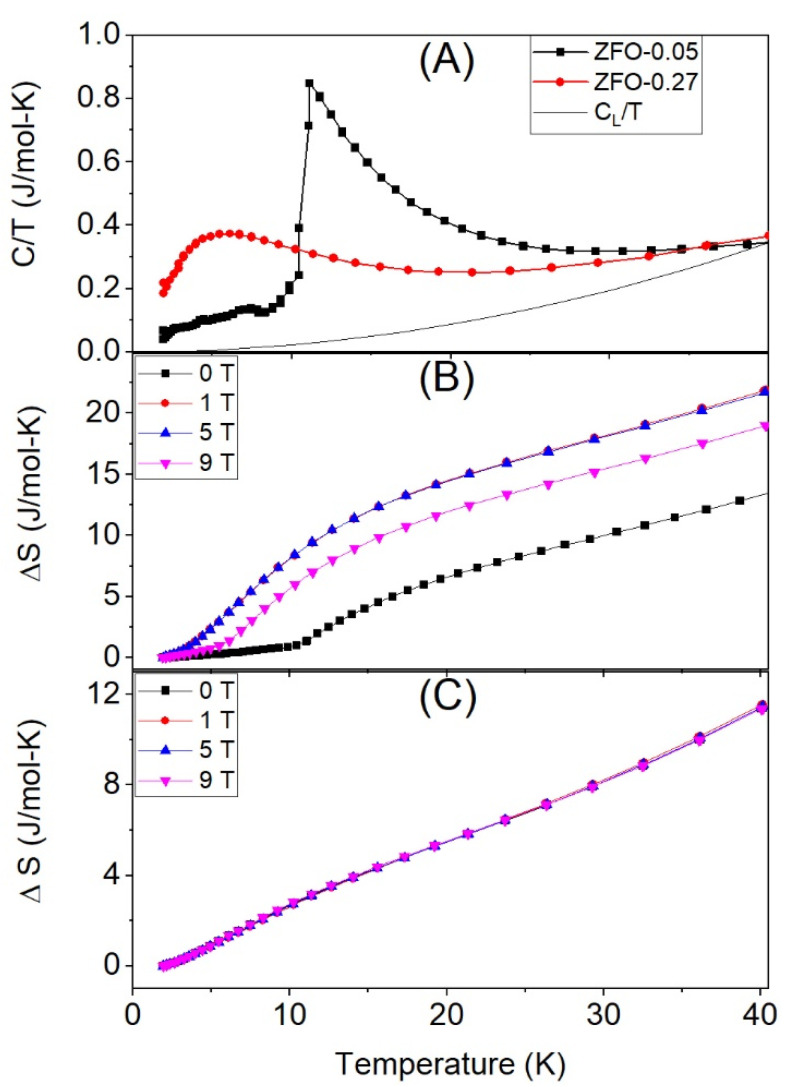
(**A**) *C/T* for sample ZFO-0.05 (black line), ZFO-0.27 (red line), and lattice contribution *C_L_/T* (continuous line); (**B**) Entropy increase at different applied magnetic fields for ZFO-0.05; (**C**) for ZFO-0.27.

**Figure 4 materials-15-01198-f004:**
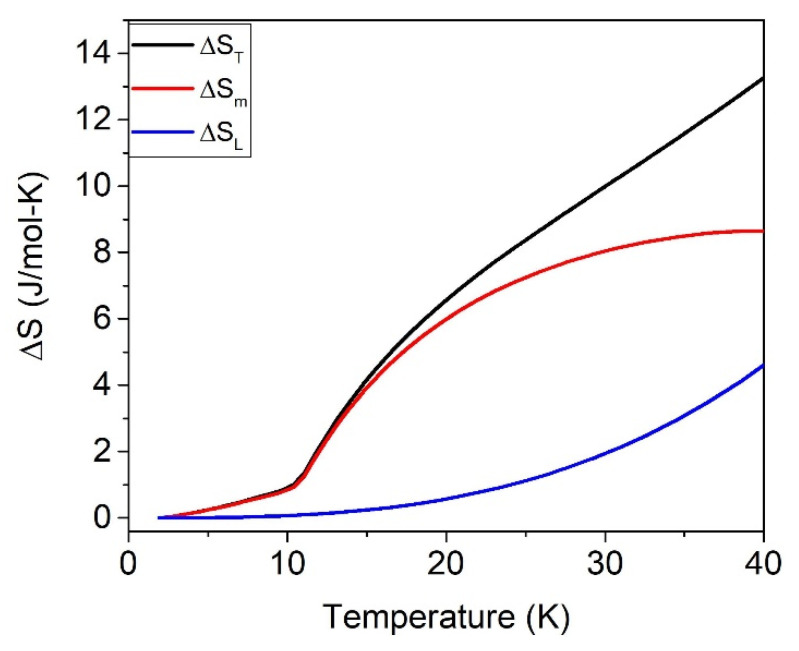
ΔS for sample with δ = 0.05, where the magnétic (red line), vibrational (blue line) and total entropy (black line) at H = 0 are shown separately.

**Figure 5 materials-15-01198-f005:**
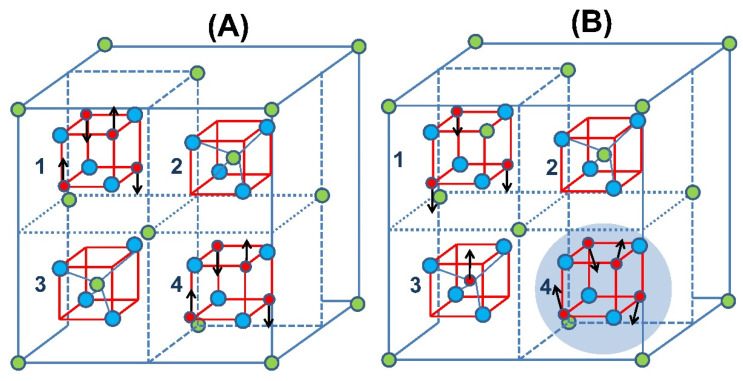
(**A**) Half-cells of ZnFe_2_O_4_ with δ = 0, Fe^3+^ (red circles), and Zn^2+^ (green circles) in their corresponding octahedral (cells 1, 4) and tetrahedral (cells 2, 3) sites. Blue circles are oxygen. The black arrows indicate the magnetic moments. (**B**) A pair of Zn-Fe cations interchanged their sites; the stronger AFM A-B super exchange interaction leads to an FM order in the B sites and promotes some kind of frustration in the first neighbor’s B sites (shadow circle).

**Figure 6 materials-15-01198-f006:**
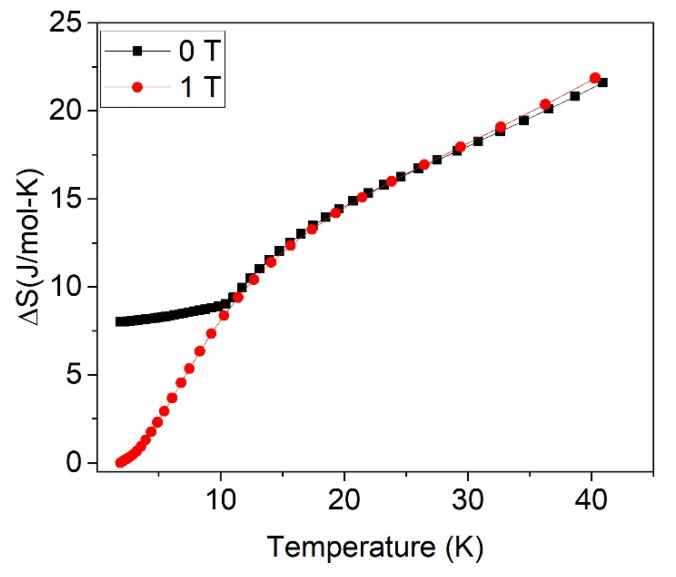
Entropy increment of ZFO-0.05 without applied field lifts up at 10 K regarding the curve at 1 T at the same temperature.

**Figure 7 materials-15-01198-f007:**
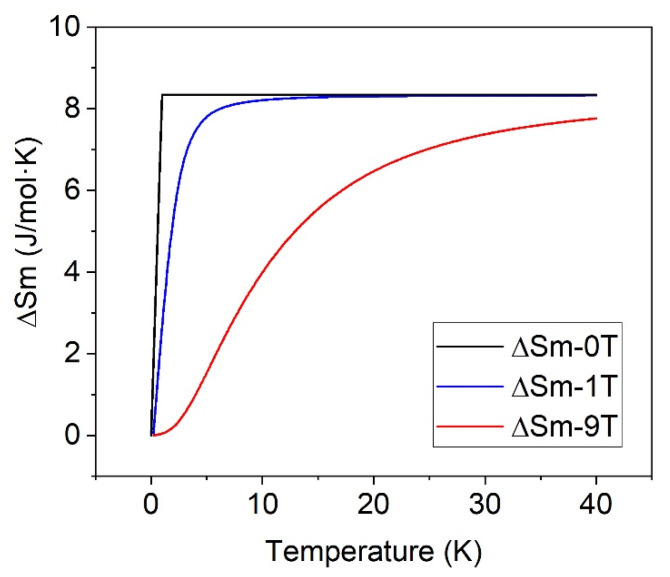
Calculated entropy increase for the paramagnetic region under a magnetic field of 0 T (black line), 1 T (blue line), and 9 T (red line).

**Figure 8 materials-15-01198-f008:**
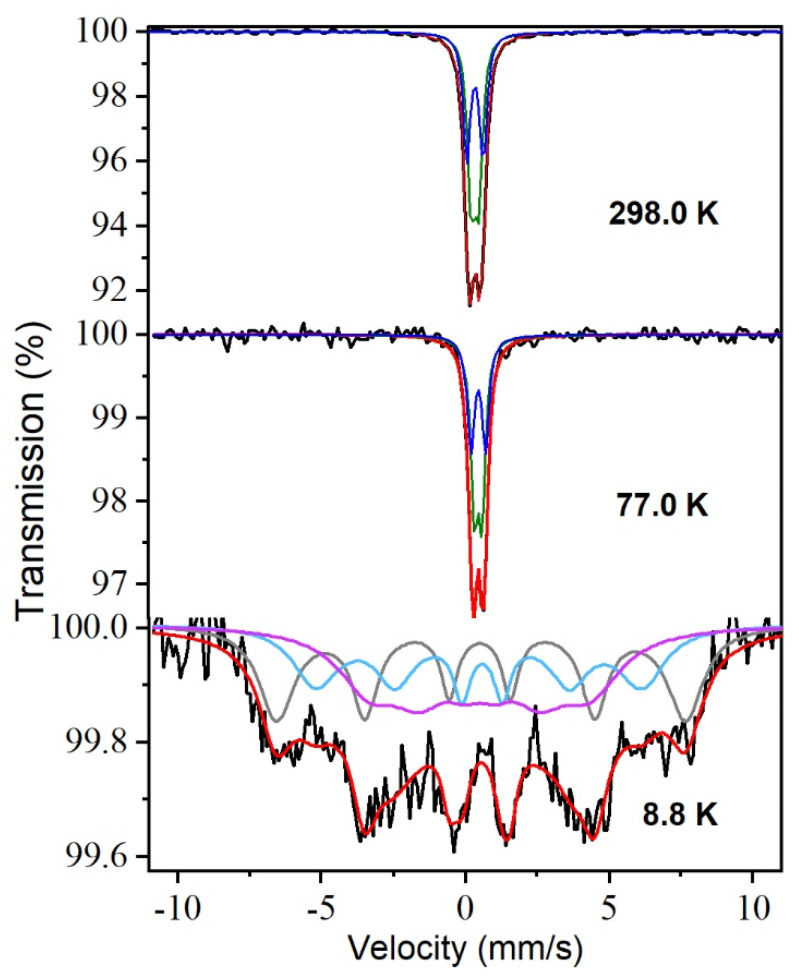
Mossbauer spectra recorded at different temperatures from δ = 0.05.

**Figure 9 materials-15-01198-f009:**
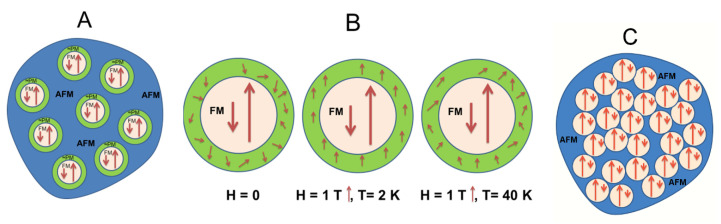
(**A**) Illustration of magnetic arrangement of ZFO-0.05 sample, white circles representing ferrimagnetic particles, the blue area is AFM converted to PM at 40 K, and the green crown represents the disordered interphase between AFM and FM regions; (**B**) Different orientation of interphase area depending on the applied field (0, 1 T), and temperature; (**C**) Magnetic arrangement of ZFO-0.27 with percolating FM clusters and blue AFM regions.

**Table 1 materials-15-01198-t001:** Microstructural parameters obtained from the Rietveld refinement of the diffraction patterns recorded using XRD and NPD.

Sample	Source	Lattice Parameter (Å)	Inversion Degree(δ)	O-Position (x = y = z)	Crystal Size (nm)	μ-Deformation (ε)
ZFO-0.05	XRD	8.4489(5)	0.05(1)	0.2416(9)	>150	-
	NPD	8.4498(5)	0.05(1)	0.2397(3)	>150	-
ZFO-0.27	XRD	8.4322(5)	0.28(2)	0.2424(5)	15(1)	0.0020(2)
	NPD	8.4373(5)	0.20(2)	0.2414(3)	14(1)	0.0019(2)

**Table 2 materials-15-01198-t002:** Total, magnetic, and lattice entropy increment (expressed in J/mol·K) for δ = 0.05 at 40 K.

*H* (*T*)	40 K		
	Δ*S_T_*	Δ*S_m_*	Δ*S_L_*
0	13.2 (1)	8.7 (1)	4.6 (1)
1–5	21.7 (1)	17.1 (1)	4.6 (1)
9	18.9 (1)	14.2 (1)	4.7 (1)

## Data Availability

Not applicable.

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
