# Peer review of "Unveiling the Hidden Entropy in ZnFe2O4"

_materials, 2022, doi:10.3390/ma15031198_

Round 1
Reviewer 1 Report
The work "Unveiling the hidden entropy in ZnFe2O4" is written relatively well and fits the topics of the journal Materials.
It is shown in the work that a {delta} value as small as 0.05 may affect 40% of the unit cells, which become locally ferrimagnetic (FiM) and coexists with AFM and spin disordered regions. The spin disorder disappears under an applied field of 1 T. Mossbauer spectroscopy confirms the presence of a volume fraction with low hyperfine field that can be ascribed to these spin disordered regions. The volume fractions of the three magnetic phases estimated from entropy and hyperfine measurements are roughly coincident and correspond to approximately 1/3 for each of them. Consequently, the so-called “hidden” entropy can be ascribed to the frustrations of the spins at the interphase between the AFM-FiM phases due to having {delta} ≈ 0 instead of ideal {delta}= 0.
For publishing this work, the following comments should be taken into account:
1. The authors introduce the concept of "hidden" entropy and cite references 19-21 but I did not see the definition. Say what "hidden" entropy means and give credible links where available. If there are no reliable sources of this concept, you must change the title of the article.
2. The authors claim that spin disorder disappears under the action of an applied field of 1 T. Figure 6 shows two graphs, respectively, at 0 and 1 Tesla, perhaps the spin order disappears at lower fields, it is necessary to provide information about other fields intermediate between 0 and 1 T.
3. Many mechanical errors, for example, in line 36 the font is different, in line 122 Fe^3+ oxidation state, the degree must be written without a dash at the bottom, please pay attention to such minor errors.
Author Response
The work "Unveiling the hidden entropy in ZnFe2O4" is written relatively well and fits the topics of the journal Materials.
It is shown in the work that a {delta} value as small as 0.05 may affect 40% of the unit cells, which become locally ferrimagnetic (FiM) and coexists with AFM and spin disordered regions. The spin disorder disappears under an applied field of 1 T. Mossbauer spectroscopy confirms the presence of a volume fraction with low hyperfine field that can be ascribed to these spin disordered regions. The volume fractions of the three magnetic phases estimated from entropy and hyperfine measurements are roughly coincident and correspond to approximately 1/3 for each of them. Consequently, the so-called “hidden” entropy can be ascribed to the frustrations of the spins at the interphase between the AFM-FiM phases due to having {delta} ≈ 0 instead of ideal {delta}= 0.
For publishing this work, the following comments should be taken into account:
- The authors introduce the concept of "hidden" entropy and cite references 19-21 but I did not see the definition. Say what "hidden" entropy means and give credible links where available. If there are no reliable sources of this concept, you must change the title of the article.
Answer: The denomination of “hidden entropy” appears in the work of Lashley et al., (PRB B 78, 104406 _2008). We have used this term in our paper and, in addition, we have added the definition as: “an entropy different from 0 at the zero point entropy”.
- The authors claim that spin disorder disappears under the action of an applied field of 1 T. Figure 6 shows two graphs, respectively, at 0 and 1 Tesla, perhaps the spin order disappears at lower fields, it is necessary to provide information about other fields intermediate between 0 and 1 T.
Answer: It is true that we have not measured the heat capacity at fields between 0 and 1T. The investigation of heat capacity as a function of the field from 0 to 1T is a time-consuming experiment and it is beyond the scope of this article. However, the proposition of the referee is considered for a future experiment.
- Many mechanical errors, for example, in line 36 the font is different, in line 122 Fe^3+ oxidation state, the degree must be written without a dash at the bottom, please pay attention to such minor errors.
Answer : The typo error has been corrected in the text paper.
Reviewer 2 Report
The article is an very interesting approach on a specific field of the alloys entropy and I believe will bring future citation, considering the presented aspects.
I have only one minor remark on the background of the study, the references should be detailed, not presented in bulk [1-4] [6-18].
The methodology and results with discussion on the article are well presented.
Author Response
The article is an very interesting approach on a specific field of the alloys entropy and I believe will bring future citation, considering the presented aspects.
I have only one minor remark on the background of the study, the references should be detailed, not presented in bulk [1-4] [6-18].
Answer: the format of the bibliography is given by journal.
The methodology and results with discussion on the article are well presented.
Reviewer 3 Report
The issue of the entropy in ferrites attracts some attention and deserves the interest of the community of materials scientists. The manuscript addresses this issue, reporting a calorimetric, structural and Moessbauer study of ZnFe2O4, emphasizing the importance of small, non-zero values of delta parameter for understanding the physical picture of the studied materials.
The manuscript presents results being of interest for the community and the topic itself is interesting. I would recommend it for publication in Materials, but provided that the Authors give prior consideration to the detailed issues listed below:
- Line 63 and other places: some symbols (like cm3) should be placed in upper index.
- Line 88: the integration limits should be included in the integral to make the expression more formal.
- Equation 1, line 147: the parameter c should be described in slightly more precise way-how it is determined (from fitting in some temperature range)? What was its value for both samples? Moreover, equation 1 seems to provide rather the vibrational contribution to specific heat CL and not the total CT – or maybe the formula is used only over 40 K to model the total specific heat, while below 40 K it provides the vibrational contribution to be subtracted from total specific heat in order to calculate the magnetic contribution? In order to illustrate the whole procedure, a plot similar to Fig. 3 in https://doi.org/10.1103/PhysRevB.52.10122 (Ref. 20) might be particularly useful.
- Line 152 and other places (for example Figure 7): Some symbols (like ΔSm) should be placed in lower indices.
- Line 175: the meaning of the parameter J should be explained and the J value should be given - probably it is the same as in Ref. 20?
- Figure 5: I feel there is some unclarity in the description of panels A and B of Figure 5 and in the designation of some lattice sites by A and B (to mention A-B exchange interaction). It should be corrected to make the whole discussion of the figure clear for the Reader. The sites A and B should be marked in explicit way in the scheme.
- The notation of functions in Equation 2 and in-line formulas should be improved.
- Citation of Ref. 19 seems to refer to the work https://doi.org/10.1088/1742-6596/145/1/012029, so the reference to J. Phys.: Conf. Ser. 145 (2009) 012029 should be given.
Author Response
The issue of the entropy in ferrites attracts some attention and deserves the interest of the community of materials scientists. The manuscript addresses this issue, reporting a calorimetric, structural and Moessbauer study of ZnFe2O4, emphasizing the importance of small, non-zero values of delta parameter for understanding the physical picture of the studied materials.
The manuscript presents results being of interest for the community and the topic itself is interesting. I would recommend it for publication in Materials, but provided that the Authors give prior consideration to the detailed issues listed below:
- Line 63 and other places: some symbols (like cm3) should be placed in upper index.
Answer: the error has been corrected in the text .
- Line 88: the integration limits should be included in the integral to make the expression more formal.
Answer: Limits 2 and 40 have been included in the integral:
- Equation 1, line 147: the parameter c should be described in slightly more precise way-how it is determined (from fitting in some temperature range)? What was its value for both samples? Moreover, equation 1 seems to provide rather the vibrational contribution to specific heat CL and not the total CT – or maybe the formula is used only over 40 K to model the total specific heat, while below 40 K it provides the vibrational contribution to be subtracted from total specific heat in order to calculate the magnetic contribution? In order to illustrate the whole procedure, a plot similar to Fig. 3 in https://doi.org/10.1103/PhysRevB.52.10122 (Ref. 20) might be particularly useful.
Answer: the equation has been rewritten as CL = 12/5*R*(T/ΘD)3 . The figure 3A shows now the contribution of the lattice to CL/T.
- Line 152 and other places (for example Figure 7): Some symbols (like ΔSm) should be placed in lower indices.
Answer: it has been corrected in the text.
- Line 175: the meaning of the parameter J should be explained and the J value should be given - probably it is the same as in Ref. 20?
Answer: The parameter J is the total quantum moment, which in the case of Fe3+ is 5/2. This has been explained in the text.
- Figure 5: I feel there is some unclarity in the description of panels A and B of Figure 5 and in the designation of some lattice sites by A and B (to mention A-B exchange interaction). It should be corrected to make the whole discussion of the figure clear for the Reader. The sites A and B should be marked in explicit way in the scheme.
Answer: The reference to octahedral and tetrahedral sites has been indicated in the figure caption.
The notation of functions in Equation 2 and in-line formulas should be improved.
Answer: The equation format has been improved.
- Citation of Ref. 19 seems to refer to the work https://doi.org/10.1088/1742-6596/145/1/012029, so the reference to J. Phys.: Conf. Ser. 145 (2009) 012029 should be given.
Answer: The correct reference has been included.
Round 2
Reviewer 1 Report
I propose to accept this work for publication.